# Triple-Camera Rectification for Depth Estimation Sensor

**DOI:** 10.3390/s24186100

**Published:** 2024-09-20

**Authors:** Minkyung Jeon, Jinhong Park, Jin-Woo Kim, Sungmin Woo

**Affiliations:** 1Department of Information and Communication Engineering, Korea University of Technology and Education (KOREATECH), Cheonan-si 31253, Republic of Korea; jun0124z@koreatech.ac.kr; 2R&D Center, VisioNext, Seongnam-si 13488, Republic of Korea; jhp.park@hanwha.com (J.P.); jinwoo.kim@hanwha.com (J.-W.K.)

**Keywords:** rectification, calibration, disparity, stereoscopic camera

## Abstract

In this study, we propose a novel rectification method for three cameras using a single image for depth estimation. Stereo rectification serves as a fundamental preprocessing step for disparity estimation in stereoscopic cameras. However, off-the-shelf depth cameras often include an additional RGB camera for creating 3D point clouds. Existing rectification methods only align two cameras, necessitating an additional rectification and remapping process to align the third camera. Moreover, these methods require multiple reference checkerboard images for calibration and aim to minimize alignment errors, but often result in rotated images when there is significant misalignment between two cameras. In contrast, the proposed method simultaneously rectifies three cameras in a single shot without unnecessary rotation. To achieve this, we designed a lab environment with checkerboard settings and obtained multiple sample images from the cameras. The optimization function, designed specifically for rectification in stereo matching, enables the simultaneous alignment of all three cameras while ensuring performance comparable to traditional methods. Experimental results with real camera samples demonstrate the benefits of the proposed method and provide a detailed analysis of unnecessary rotations in the rectified images.

## 1. Introduction

Depth estimation using stereo cameras involves estimating distance by analyzing the disparity between left and right images, a process essential for applications like the autonomous driving of robots or cars and object detection. Traditionally, the Semi-Global Matching (SGM) method, known for its speed and accuracy, has been widely used [1,2,3,4]. Recently, with the advancement of on-device AI technology, research on disparity estimation using neural networks has become increasingly active [5,6,7,8,9,10,11]. The core challenge of stereo matching lies in identifying the corresponding point in the right (R) image that matches a point in the left (L) image.

Image rectification is a technique that places L and R images on the same 3D plane and aligns matching points horizontally, simplifying the stereo matching problem from a 2D search to a 1D search. This technique is crucial for real-time performance in stereo camera matching, which demands high frame rates. However, rectification is challenging due to ambiguities in feature matching, lens distortion, sensitivity to initial conditions, and other factors. Notably, the rectification problem does not have a unique solution, leading to the development of various methods tailored to specific needs.

One solution to facilitate the detection of reference points for feature matching is to use images taken with a checkerboard chart, consisting of squares arranged in a pattern that is easily detectable within the camera’s field of view. Twists and misalignments often occur during the manufacturing assembly process for various reasons. Based on the known 3D pattern information, calibration to obtain intrinsic parameters such as focal length, camera center, and lens distortion is performed first. This is followed by rectification to find the common intrinsic parameters and rotation matrix for both cameras. Once the output parameters of the calibration and rectification are recorded in permanent memory, the captured images from the L and R cameras can be remapped on the fly.

Bouguet’s method [12], provided as a function in image processing tools such as MATLAB and OpenCV, is the most popular rectification algorithm. It rotates and scales the left and right images to align detected features on the same horizontal line. In most cases, the resulting images are well aligned, making it easy to compare reference points between the two images. However, while this method is suitable for two cameras with very different fields of view, it is not ideal for rectifying stereo cameras physically designed in parallel. Figure 1 shows a before-and-after rectification of L and R of a stereo camera using Bouguet’s method, where the resulting images tend to be excessively rotated compared to the original. This unintentional rotation can be problematic in applications such as autonomous driving, where the position or orientation of the camera apparatus is critical.

Recently, depth estimation cameras have started incorporating a third camera alongside the stereo pair. This additional camera, equipped with a filter that absorbs specific wavelengths, adds color information to the disparity map, which is then used to create a 3D point cloud [13]. This setup is also beneficial for multi-view stereo vision or high-precision 3D scanning systems, as it provides a broader field of view. However, there can be cases where the third camera and the depth information are not aligned, leading to poor 3D quality. This misalignment occurs because existing alignment methods do not consider the third camera, necessitating an additional calibration and rectification process to align the third camera with the stereo cameras.

In this paper, we introduce a novel rectification technique that simultaneously aligns the left, right, and third cameras without causing unintended image rotation, using just a single captured image. Unlike existing methods that necessitate multiple checkerboard images, our approach aligns and remaps all three cameras in one unified process. This method offers a significant advantage by eliminating the need for separate rectification and remapping for the third camera. Moreover, the rectified image used for disparity creation remains unrotated, ensuring a stable display. The proposed technique achieves stereo-matching performance comparable to Bouguet’s algorithm, even when aligning three cameras simultaneously, as evidenced by tests on 19 camera samples from the depth camera manufacturing process. The aligned images demonstrate superior disparity matching quality. This system is highly effective in correcting mechanical alignment errors in depth cameras and can significantly improve the accuracy of disparity estimation using deep learning.

We review previous work in Section 2, introduce the proposed rectification method in Section 3, and provide experimental results and analysis in Section 4. Further discussion is provided in Section 5. Section 6 concludes the paper.

## 2. Related Work

The Hartley [14] and the Bouguet rectifications [12] are widely recognized for their simplicity and performance, making them popular choices in image-processing toolkits like MATLAB and OpenCV. The Hartley algorithm focuses on aligning the epipolar lines of two images by finding and matching corresponding features between the left and right images. Although straightforward, it cannot accurately determine the size or distance of objects because it does not account for the camera’s intrinsic parameters.

In contrast, Bouguet’s method [12] is effective when camera calibration parameters are known. It aims to minimize reprojection distortion while maximizing the common viewing area. This method divides the rotation matrix into two separate matrices for the left and right cameras. Each camera rotates half a rotation to align their principal rays parallel to the vector sum of their original directions, achieving coplanar alignment but not row alignment. To horizontally align the epipolar lines, a rotation matrix is computed starting from the epipole’s direction, which is along the translation vector between the two cameras’ centers of projection.

Monasse [15] introduced a three-step image rectification method that improves stereo rectification by minimizing distortion through geometric transformations. This process involves aligning epipolar lines to make them parallel and horizontal, using a robust algorithm that splits the camera rotation into three steps, thereby reducing computational complexity and improving accuracy. Unlike traditional methods, it minimizes a single parameter, the focal length, making it more resilient to initial epipolar line misalignment.

Kang [16] presented a stereo image rectification method using a horizontal baseline to address geometric errors caused by manual camera arrangements and internal camera characteristic differences. Unlike traditional calibration-based methods that often result in visual distortions like image skewness, this method calculates a baseline parallel to the real-world horizontal line, estimates camera parameters, and applies a rectification transform to the original images to produce rectified stereo images with minimal distortion.

Isgrb [17] proposed a novel algorithm for projective rectification that bypasses the explicit computation of epipolar geometry or the fundamental matrix. Instead, it leverages the known form of the fundamental matrix for rectified image pairs to establish a minimization problem that directly yields the rectifying homographies from image correspondences. This approach simplifies the correspondence problem by aligning corresponding points along horizontal scanlines, thus reducing the problem from 2D to 1D.

Pollefeys [18] introduced a rectification method for stereo images capable of handling all possible camera motions. This method employs a polar parameterization of the image around the epipole, transferring information between images through the fundamental matrix. This approach avoids the issues encountered by traditional and cylindrical rectification methods, which can result in excessively large images or fail to rectify altogether.

Fusiello [19] presented a linear rectification method for general, unconstrained stereo rigs, which computes rectifying projection matrices from the original camera’s perspective projection matrices. This method computes rectifying projection matrices from the original camera’s perspective projection matrices, providing a compact and easily reproducible solution.

Lafiosca [20] introduced a rectification method to compute a homography that reduces perspective distortion, offering an improvement over the rectification method by Loop and Zhang [21]. This approach provides a closed-form solution that avoids the need for numerical optimization.

While existing methods are general-purpose algorithms designed to align images taken from various angles and primarily focus on reducing alignment errors, the proposed method is specifically tailored for mass-producing stereo cameras. It simultaneously aligns three cameras, ensuring that the reference image is not unintentionally rotated. To facilitate this process, we also propose an image-capturing environment that includes a specially designed checkerboard and its specific configuration.

## 3. Proposed Method

Figure 2 visualizes the existing calibration and rectification methods for aligning the left, right, and RGB cameras, as well as the proposed triple-camera alignment system. In existing systems, depth estimation typically involves the left and right cameras, with an additional RGB camera for generating the 3D point cloud. These systems require two separate calibration and rectification processes: one for estimating depth using the left and right cameras, and another for encoding color information using the left and RGB cameras. The rectification outputs a rotation matrix in a common 3D plane, considering the relative relationship between the two cameras. Although the left camera is common to both rectification processes, the rotation matrix and common plane for each are different. Additionally, the existing rectification process is time-consuming because multiple checkerboard images are required to extract more features and determine the intrinsic and extrinsic parameters of the cameras. In contrast, the proposed method uses a single image for each camera and requires only one unified calibration and rectification step, where both the right and RGB cameras are calibrated and rectified relative to the left camera, which serves as the main reference for depth estimation and 3D point-cloud generation.

### 3.1. Checkerboard Design and Lab Setting

To perform camera calibration and rectification, it is necessary to capture multiple chart images from different positions to detect feature points. The proposed method utilizes four checkerboards for a single shot, as illustrated in Figure 3A, each placed in a different quadrant. Each checkerboard consists of a 19 × 14 grid pattern, with squares measuring 24 mm on each side. The top-left checkerboard faces straight ahead, while the other three are rotated approximately 30 degrees around different axes, facilitating the generation of 3D multi-view geometry points. Each checkerboard is positioned to include as much of the corner areas of the image as possible, ensuring that distortion correction and rectification are effectively applied to the edges of the image. With this setup, a total of 228 × 4 = 912 reference points are obtained.

### 3.2. Checkerboard Corner Detection

The captured image in Section 3.1 is divided into four sections to utilize the existing corner-detection methods, and corner detection is performed in each section. To facilitate this, the remaining area except the approximate chart area considering the margin is first masked in black as described in Figure 3B, and then corner detection is performed. Various feature detection methods such as Harris corner detection [22], SIFT [23], SURF [24], ORB [25], FAST [26], BRIEF [27], HOG [28], and MSER [29] often show poor performance near the out-of-focus area (in our case, it mostly happens in the off-center regions), due to lens-assembly defects that often occur in the early stage of manufacturing. We adopt Geiger’s method [30], which accurately and reliably detects corner locations even in chart images with distorted or blurred areas. The results of detecting corners in the checkerboard image using the proposed method are shown in Figure 4. The red dots in each grid pattern represent the corners detected in the left image, and the same corner-detection process is applied to the right and RGB images.

### 3.3. Calibration and Rectification

For triple-camera rectification, the process begins with single camera calibration for the left, right, and RGB images to determine each camera’s intrinsic/extrinsic parameters and lens-distortion coefficients. This involves obtaining extrinsic parameters, which represent the transformation from the 3D world to the camera coordinate system for the checkerboard reference point, and intrinsic parameters, which map 3D camera coordinates to 2D image coordinates [31,32]. Figure 5 illustrates the coordinates and transformation matrices in 2D and 3D during the rectification process, depicting the transition from observed to rectified points for a single camera.

#### 3.3.1. Single-Camera Calibration

The calibration process involves determining the camera matrix M, called an intrinsic matrix and extrinsic matrix E. The intrinsic matrix M comprises the focal lengths $f_{x}$ and $f_{y}$ along the x and y axes, and the principal points $c_{x}$ and $c_{y}$, as depicted in Equation (1).
(1)$$M=\left[\begin{array}{ccc} f_{x} & 0 & c_{x}\\ 0 & f_{y} & c_{y}\\ 0 & 0 & 1 \end{array}\right]$$

The extrinsic matrix E consists of rotation and translation parameters that transform the 3D corners in world coordinates into the 3D coordinates of the camera, as shown in Equation (2). Since the proposed system uses four checkerboards, there are four extrinsic matrices ($E_{1}$, $E_{2}$, $E_{3}$, $E_{4}$). Each external matrix corresponding a single checkerboard consists of a 3 × 3 rotation matrix R and a 3 × 1 translation matrix T along the three axes.
(2)$$E=\left[R|T\right]$$

In Figure 5, q and Q represent the corner points in the 2D raw image and the 3D raw plane, respectively. Additionally, $q^{\prime }$ and $Q^{\prime }$ are the 2D and 3D coordinates after lens-distortion correction, and $q^{\prime \prime }$ and $Q^{\prime \prime }$ are the corresponding coordinates after rectification.
$$q_{x}=q_{x}^{\prime }+\left(q_{x}^{\prime }-c_{x}\right)\left[k_{1}r^{2}+k_{2}r^{4}\right]$$
(3)$$\;q_{y}=q_{y}^{\prime }+\left(q_{y}^{\prime }-c_{y}\right)\left[k_{1}r^{2}+k_{2}r^{4}\right]$$

Equation (3) describes the lens-distortion model, where ($q_{x}$, $q_{y}$) and ($q_{x}^{\prime }$, $q_{y}^{\prime }$) are the coordinates of the corner points in the distorted (observed) and lens-corrected image, respectively [31]. $k_{1}$ and $k_{2}$ in Equation (3) represent the first and second lens-distortion coefficients. The distance $r=\sqrt{{q_{x}^{\prime }}^{2}+{q_{y}^{\prime }}^{2}}$ between the principal point and the distortion-corrected coordinates determines the degree of radial distortion.

$c_{x}$ and $c_{y}$ from the 3D to 2D transformation matrix M are used for both the distorted and corrected images, as shown in Equation (3). Likewise, the conversion from 2D image coordinates to 3D homogenous coordinates is achieved by multiplying $M^{-1}$ to q and $q^{\prime }$. Given the distorted image points q and the corresponding 3D corner points in world coordinates, obtaining the undistorted function, Undist() in Figure 6 is an inverse problem. This can typically be estimated using iterative methods or approximate algorithms, such as the Newton–Raphson method [33] or numerical optimization [34].

In the calibration process, the intrinsic matrix M is obtained first by singular value decomposition (SVD). The initial extrinsic matrix E is subsequently derived [31]. The 3D world coordinates of checkboard corners $Q_{i}^{W}$ are transformed to camera 3D coordinate $Q^{C}$ by multiplying E. The initial $k_{1}$ and $k_{2}$ are then obtained using Equation (3). Once the initial parameters are established, optimization is performed to minimize the Euclidean error between the observed corner points $q_{ij}$ and the estimated corner points $\hat{q}$ using Equation (4).
(4)$$\underset{M,\;E_{1}\sim E_{4},k_{1},\;k_{2}\;}{\mathrm{argmin}}\sum_{j}^{N}\sum_{i}^{K}\|q_{ij}-{\hat{q}}_{ij}\;\left(M,\;E_{1},E_{2},\;E_{3},E_{4},k_{1},\;k_{2},\;Q_{i}^{W}\right){\|}^{2}\;$$
where N and K are the number of checkerboards in the chart image and the total number of corner points in each checkerboard, respectively (N = 4, K = 228). While the intrinsic matrix M is unaffected by external environmental factors, $E_{1\sim 4}$ vary depending on the camera or checkboard position. Thus, E is required only for the optimization process and is not documented for calibration purposes. The 3D points in world coordinates $Q_{i}^{W}$ depend on each checkerboard’s locations. However, even if we assume that the world coordinates of the four checkerboards are the same, as long as only $E_{1\sim 4}$ changes, it is acceptable to use $Q_{i}^{W}$ instead of $Q_{ij}^{W}$. In Equation (4), the intrinsic/extrinsic matrices and lens-distortion coefficients are optimized to minimize the difference between the observed points $q_{ij}$ and estimated points in 2D image coordinates.

#### 3.3.2. Triple-Camera Calibration

Once the intrinsic and extrinsic parameters and the lens-distortion coefficients for each camera are obtained, the relative relationship of the three cameras is considered. As mentioned in [12], the relative rotation $R^{R}$ and translation $T^{R}$ of the right to the left camera are expressed in Equation (5). Similarly, the relative rotation $R^{RGB}$ and translation $T^{RGB}$ of the RGB to the left camera are expressed in Equation (6).
$$R^{R}=R_{j}^{R}{\left(R_{j}^{L}\right)}^{T}$$
(5)$$T^{R}=T_{j}^{R}-R^{R}T_{j}^{L}$$
$$R^{RGB}=R_{j}^{RGB}{\left(R_{j}^{L}\right)}^{T}$$
(6)$$T^{RGB}=T_{j}^{RGB}-R^{RGB}T_{j}^{L}$$
where $R_{j}^{L}$, $R_{j}^{R}$, and $R_{j}^{RGB}$ are the left, right and RGB camera’s external rotation matrices, respectively, and $T_{j}^{L}$, $T_{j}^{R}$, and $T_{j}^{RGB}$ are the translation matrices. Although the relative rotation and translation matrices should theoretically be the same regardless of the checkerboard location, slight variations occur. A single set of $R^{R}$, $R^{RGB}$, $T^{R}$, and $T^{RGB}$ is determined, and the intrinsic matrices of each camera are adjusted for consistency. The loss function in Equation (7) minimizes the difference between the observed and the projected 2D points in each camera using Equations (5) and (6). For the extrinsic matrices in Equation (7), only $E_{1\sim 4}^{L}$ are optimized, while the extrinsic matrices of the right camera and the RGB camera are computed based on the relative rotation and translation, as described in Equation (8).
(7)$$\underset{M^{c},k_{1}^{c},k_{2}^{c},E_{1\sim 4}^{L},R^{R},R^{RGB},T^{R},T^{RGB}\;}{\mathrm{argmin}}\sum_{c}\sum_{j}^{N}\sum_{i}^{K}\|q_{ij}^{c}-{\hat{q}}_{ij}^{c}\;\left(M^{c},E_{1\sim 4}^{c},k_{1}^{c},k_{2}^{c},Q_{i}^{W}\right){\|}^{2}\;\;$$
$$c\in \left\{L,R,\;RGB\right\}$$
(8)$$E_{j}^{R}=\left[R_{j}^{R}\right|T_{j}^{R}],R_{j}^{R}=R^{R}R_{j}^{L},\;T_{j}^{R}=T^{R}+R^{R}T_{j}^{T}$$

For initial values of these relative matrices, median matrix values across all checkboards are used.

#### 3.3.3. Triple-Camera Rectification

The triple-camera calibration performed in the previous section does not ensure that the epipolar lines of all three cameras are horizontally aligned, indicating that the image planes of the three cameras in 3D space are not equivalent. Rectification aims to find a common intrinsic matrix $M_{rec}$ and rotation matrices $R_{rec}^{c}$ for each camera, creating a common plane for all and making the epipolar lines parallel. To achieve this, the corrected coordinates $q^{\prime }$ of the observed points in 2D image space q is first estimated using the Undist() function mentioned in Section 3.3.1. The distortion-corrected point $q^{\prime }$ is transformed to 3D coordinates $Q^{\prime }$ by multiplying the inverse of $M^{c}$, and then rotated again by $R_{rec}^{c}$ to become rectified points $Q^{\prime \prime }$ in 3D camera coordinates. Lastly, distortion-free and rectified $q^{\prime \prime }$ is obtained by multiplying the common intrinsic matrix $M_{rec}$. In summary, the rectified 2D image point $q^{\prime \prime }$ from the observed point q in a camera is calculated as follows:(9)$$q^{\prime \prime }=M_{rec}R_{rec}\cdot Undist\left(M^{-1}q\right)$$

The resulting rotation matrix $R_{rec}^{c}$ for each camera necessarily rotates the original image to place 3D points in a common plane. However, there can be multiple common planes to satisfy the condition. The misalignment in the manufacturing process among different cameras is not usually severe. Disparity computation is performed in various ways from the rectified images, and the resulting disparity map should be visible to the external display. The rectified, randomly rotated field of view in the disparity map is not only unsuitable for visual display, but also inadequate for distance estimation in autonomous driving. Therefore, the proposed method is designed to minimize the vertical location error of the rectified points subjected to $R_{rec}^{L}=I$ as follows:(10)$$\underset{M_{rec},\;R_{rec}^{R},\;R_{rec}^{RGB}}{\mathrm{argmin}}\left[\begin{array}{c} \sum_{j}^{N}\sum_{i}^{K}\|{q_{y}^{\prime \prime }}_{ij}^{L}-{q_{y}^{\prime \prime }}_{ij}^{R}{\|}^{2}\;\\ +\sum_{j}^{N}\sum_{i}^{K}\|{q_{y}^{\prime \prime }}_{ij}^{L}-{q_{y}^{\prime \prime }}_{ij}^{RGB}{\|}^{2}\; \end{array}\right]$$
$$\mathrm{subjected}\;\mathrm{to}\;R_{rec}^{L}=I\;\mathrm{and}\;\frac{f_{x,rec}}{f_{x}^{L}}>\gamma$$
where ${q_{y}^{\prime \prime }}_{ij}^{c}$ represents the y-coordinate of the rectified 2D point $q^{\prime \prime }$ in Equation (9) for each camera. γ is a minimum threshold value for the ratio of the focal length in the rectified image, $f_{x,rec}$ to the focal length of the left camera $f_{x}^{L}$. In the process of finding $M_{rec}$, $R_{rec}^{R}$, $R_{rec}^{RGB}$ that minimize Equation (10), $R_{rec}^{L}$ is fixed as an identity matrix. Consequently, if there is significant misalignment among three cameras, reducing $f_{x,rec}$ to decrease the size of the rectified image naturally helps minimize the vertical difference between rectified points. However, since reducing the image size is not desirable, a threshold value γ is used to ensure a minimum size for $f_{x,rec}$, thereby preventing the rectified image from being excessively reduced. Once $M_{rec}$, $R_{rec}^{R}$, $R_{rec}^{RGB}$ are obtained, the relationship between q and $q^{\prime \prime }$ is established and stored as a mapping table for each camera. This mapping is then used to remap q and $q^{\prime \prime }$ for each frame, resulting in a rectified image. The optimizations in Equations (4), (7) and (10) uses the Levenberg–Marquardt (LM) algorithm [35]. This method combines the Gradient descent method and the Gaussian–Newton method, offering both speed and stability. The Trust-Region-Reflective algorithm [36] can also be selected as an optimization method. We present a summary of the proposed algorithm below as Algorithm 1, along with an example illustrating the estimated parameters and errors produced at each step.
**Algorithm 1.** Require: three checkerboard images captured by cameras arranged in a row [30,31].
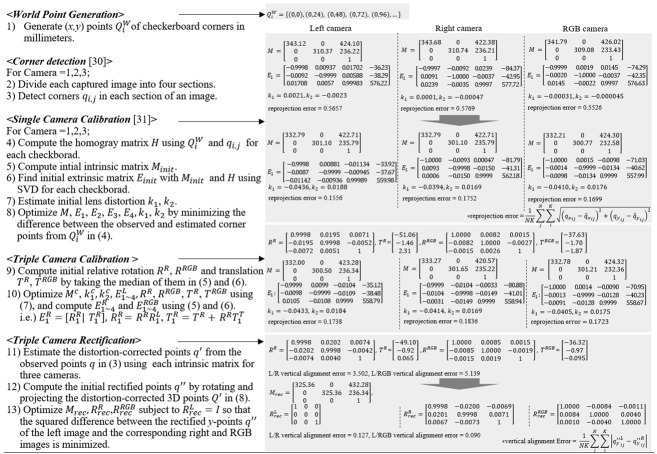


## 4. Experimental Results

To evaluate the performance of the proposed method, a newly designed stereoscopic camera with a wide field of view was constructed, as shown in Figure 6A. During the manufacturing process, 19 pairs of left, right, and RGB raw images with a resolution of 1280 × 800 were captured from 19 different camera modules. These images were used to capture the displayed checkerboard pattern. The left/right images are in 8-bit format, and the RGB images are in 24-bit JPG format. During the calibration and rectification process, the images are resized to an output size of 848 × 480 to optimize real-time performance in stereo matching. The evaluation metric for the vertical alignment error between two rectified cameras *A* and *B* is based on the Mean Absolute Error (MAE), calculated as follows:(11)$$err_{v}\left(A,\;B\right)=\frac{1}{NK}\sum_{j}^{N}\sum_{i}^{K}\left|{q_{y}^{\prime \prime }}_{ij}^{A}-{q_{y}^{\prime \prime }}_{ij}^{B}\right|\;$$

Table 1 shows $err_{v}\left(L,\;R\right)$ and $err_{v}\left(L,\;RGB\right)\;$ from the rectified images for the existing methods and the proposed method, across 19 samples. All methods used the same corner-detection results before calibration, and for the Fusiello [19] and Lafiosca [20] methods, which do not provide calibration, we used the intrinsic parameters from the triple-calibration results of the proposed method. All comparison methods were tested separately for the L/R and L/RGB cases, whereas only the proposed method performed the simultaneous calibration and rectification of L/R/RGB. γ of 0.98 was used in Equation (10) unless otherwise specified. The proposed method achieved an average alignment error of 0.112 for L/R and 0.085 for L/RGB, ranking third among all compared methods. Bouguet and Lafiosca’s methods, which are based on calibrated scenarios, achieved the best scores, while Hartley’s method, designed for an uncalibrated scenario, ranked fourth. Fusiello’s method, although relatively simple and aimed at calibrated cameras, lacks lens-distortion modeling, resulting in the worst performance.

To scrutinize the rectification results in Table 1, we listed the relative angles and locations among the L/R/RGB cameras from the triple-camera calibration in Table 2. Since the RGB camera images are converted to grayscale before processing, it is assumed that there are no distinct differences between L/R and L/RGB rectification, except for the baseline, which reflects the relative *y* locations in Table 2. In all methods, the average error for L/RGB in Table 1 is approximately 30% lower than that for L/R, which can be attributed to the difference in baselines. This is because, for camera pairs with the same angular deviation around the XYZ axes, a longer baseline makes the rectification process more sensitive to rotations around these axes. Any small misalignment in rotation can cause significant misalignment in the rectified images, increasing the overall error.

The goal of rectification is to simplify stereo matching by reducing a 2D search to a 1D search. To evaluate how the error difference between Bouguet’s method and the proposed method affects stereo-matching results, we computed the disparity using the SGM method [1] on left and right images before and after rectification. Figure 7 compares the resulting depth maps from 19 camera samples. The first to third columns show the original left images, the corresponding rectified images from Bouguet’s method, and the rectified images from the proposed method, respectively. The fourth to last columns display the disparity map without rectification, the disparity maps obtained using Bouguet’s method, and those generated using the proposed method. The disparity maps from both Bouguet’s and the proposed rectified images are quite similar, with both enhancing the disparity map to a level where the 3D geometry of the calibration chart can be reliably estimated. However, the rectified left images from Bouguet’s method in the second column show random rotations of the reference checkerboards, which were vertical in the original images, whereas the proposed method preserves the original checkerboard orientation in all rectified images. This tendency is also evident in the disparity maps in the fifth and sixth columns of Figure 7.

To determine the extent of undesired perspective rotation in the rectified images, we obtained the positions of the three red-marked points on the front-facing chart in the top left corner of Figure 3A and calculated the horizontal and vertical angle rotations after rectification as indicated by the red arrows. Figure 8, Figure 9 and Figure 10 illustrate the results for the left, right, and RGB images for each method. Existing methods, such as [12,19,20], exhibit significant rotations from the original images, with a maximum deviation of approximately 3 degrees in both horizontal and vertical directions, regardless of whether the images are left, right, or RGB. In contrast, the proposed method shows almost no rotation for the reference left image, with a maximum deviation of only 0.17 degrees, which is likely due to the scaling of the rectified image. For the right and RGB-rectified images in the proposed method, there is an average rotation of about 0.4 degrees, which is still significantly lower compared to the existing methods.

The perspective rotation in the rectification process appears to mainly result from the vertical misalignment of the stereo cameras. The samples with severe rotations have relatively large vertical misalignments of more than 1 mm (*y*-axis), as indicated in Table 2: modules 1, 8, 10, 11, 13, 15, 17, and 18 for L/R cameras, and modules 6, 9, 15, 16, and 18 for L/RGB cameras. Since the alignment error is correlated with the baseline, it is observed that a physical vertical misalignment between two closely positioned cameras can also lead to significant rotations in the rectified images, resulting in a similar vertical alignment between L/R and L/RGB, but with a larger angle of rotation in L/RGB than in L/R. If existing methods are used, the modules with larger rotation images may need to be discarded during the manufacturing process due to the substantial perspective rotations observed after rectification. Moreover, the resulting left images from L/R and L/RGB rectification in the existing methods are independent, and therefore cannot be used simultaneously, as in the proposed method. Existing methods focus on minimizing the alignment error of stereo pairs. In contrast, the proposed algorithm rectifies all three cameras simultaneously while keeping the reference camera’s movement fixed, permitting minor alignment compromises as long as they do not impact stereo matching. Consequently, the performance gap between Bouguet’s method and the proposed method in Table 1 appears to stem from the reduced degrees of freedom caused by fixing the left camera’s rotation and the concurrent rectification of the third camera.

Figure 11, Figure 12, Figure 13 and Figure 14 provide a comparison of the results before and after rectification for real-world scenes. The first row shows the original images from the left, right, and RGB cameras, while the second row offers an enlarged view of the dotted box area in each image. The third row presents the rectified images after applying the proposed method, and the fourth row displays the matching points identified using SURF. Note that all images are shown without noise reduction or other ISP tuning, leading to noticeable noise in all images and a purple tint in the RGB images. Before rectification, the images are not vertically aligned, but after rectification, the left, right, and RGB images are nearly perfectly aligned, with no unwanted rotation in the left image, which serves as the reference for disparity estimation throughout the study. The alignment errors for each process and the number of corners used in the calculation are detailed in the figure captions.

## 5. Discussion

### 5.1. Rectification with Multiple Images

Although the proposed method uses a single image for calibration and rectification, this image includes four checkerboard sub-images. Theoretically, using four separate images, each containing a single checkerboard, would not impact the accuracy of rectification. However, if each checkerboard is positioned arbitrarily and overlaps in areas of the viewing angle, it could affect the accuracy scores. To assess the accuracy of rectification with multiple captures, we created three different capturing environments, as illustrated in Figure 15, and compared them with the single-capture scenario. Table 3 summarizes the vertical alignment errors across different settings. Multi-I achieved the highest accuracy in both L/R and L/RGB rectifications, followed by Multi-II for L/R, and then the single-image rectification. Multi-III resulted in the highest error. This variation is primarily due to the coverage of the four checkerboards; overlapping checkerboard locations, as seen in Multi-I and Multi-II, help to minimize errors, while dispersed placements, as in Multi-III, lead to increased alignment errors. We also compared the rectification accuracy for the real-world scenes in Figure 11, Figure 12, Figure 13 and Figure 14 using SURF between the single-capture and Multi-I scenarios, as summarized in Table 4. However, assuming that using SURF to detect feature points in real-world images is numerically less accurate than corner detection on the checkerboard, no significant differences were observed, and both methods were able to find stable matching points.

### 5.2. The Affect of γ in Optimization

Unlike existing methods, where the left and right images are freely rotated to align the epipolar lines horizontally, the proposed method only rotates and scales the right or RGB cameras, keeping the rotation matrix of the left camera fixed as the identity matrix. Consequently, the degrees of freedom in the optimization process described in Equation (10) are reduced compared to cases where no additional constraints are applied. In general, as the focal length of the rectified image decreases, the image size also reduces, leading to a decrease in the vertical alignment error between the two cameras. An incorrect choice of γ value could prevent the optimization from converging and impact the accuracy of the rectification. Therefore, we tested the alignment error and $f_{x,rec}$ with varying γ values in (10). Figure 16 summaries the results of vertical alignment errors and focal length changes. As shown in Figure 16A,B, with the exception of module 12, most of the modules exhibit a decrease in alignment error as γ becomes lowered. This is because the focal length of the rectified images becomes smaller, as seen in Figure 16C. Smaller focal lengths result in smaller image sizes, which is advantageous for calculating vertical errors. Therefore, in the optimization of (10), the process tends to reduce the image size down to the lower limit of the focal length to minimize errors. Module 12, however, has a very small relational location in the *y*-axis for both L/R and L/RGB, as shown in Table 2, and also exhibits minimal angle differences. This indicates that the optimization for module 12 was primarily carried out using the rotation matrices,$\;R_{rec}^{R},\;R_{rec}^{RGB},\;$ with little to no change due to the focal length constraint.

## 6. Conclusions and Future Work

In this paper, we propose a method for simultaneously rectifying left, right, and RGB cameras. We comprehensively outline the entire rectification process of the proposed method using the intrinsic/extrinsic calibration and lens-distortion correction formulas. Unlike existing methods that require multiple images to rectify two cameras, our approach enables the simultaneous rectification of three camera images with a single shot, ensuring that the images do not undergo unnecessary rotation after rectification. This advantage makes the proposed technique highly useful for the production of stereo cameras for distance estimation. In our experiment, we showed that the vertical misalignment between two cameras plays a key role in perspective rotation images in the rectification. Therefore, including the camera’s translation in the optimization process for rectification is anticipated to be a topic for future research.

## Figures and Tables

**Figure 1 sensors-24-06100-f001:**
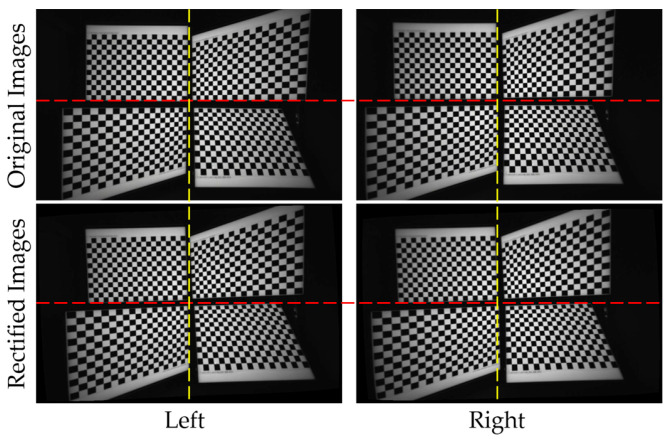
Results of Bouguet’s rectification for L and R images. (**Top**) Before rectification, (**Bottom**) after rectification, (**Left**) Images from left camera, (**Right**) Images from right camera. Please note that, although the resulting images are well-rectified, they still exhibit rotation even with minor L/R misalignment inputs.

**Figure 2 sensors-24-06100-f002:**
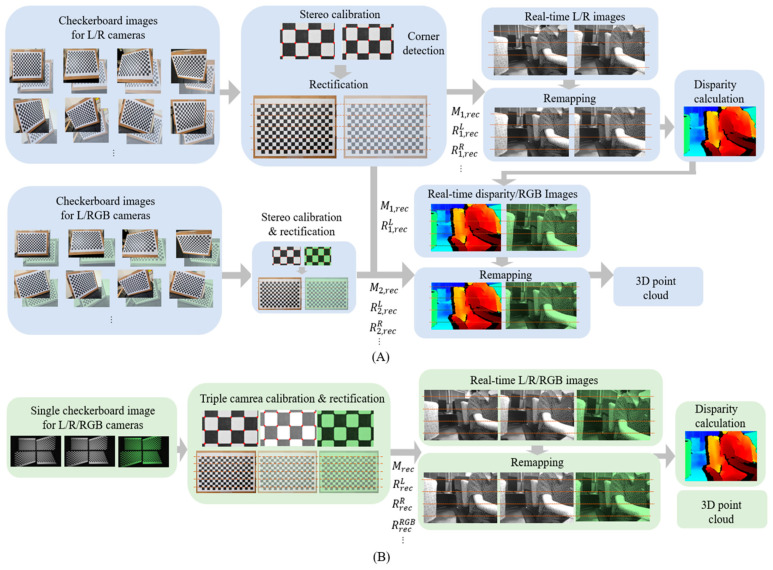
(**A**) A diagram of the existing calibration and rectification system based on stereo cameras and (**B**) the proposed simultaneous triple-camera calibration and rectification system.

**Figure 3 sensors-24-06100-f003:**
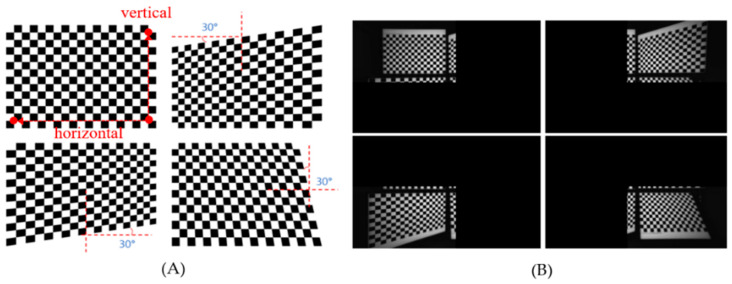
(**A**) Checkerboard configuration designed for single-shot calibration and rectification for three cameras. (**B**) Four segmented masked images for corner detection.

**Figure 4 sensors-24-06100-f004:**
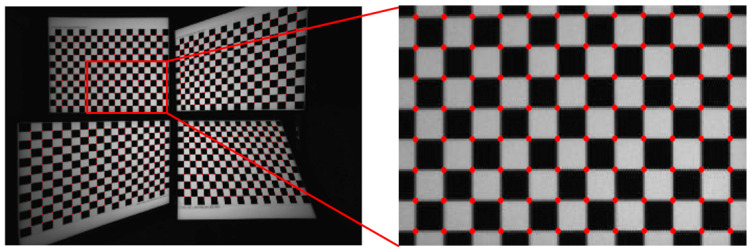
Example of corner-detection results for a checkerboard image.

**Figure 5 sensors-24-06100-f005:**
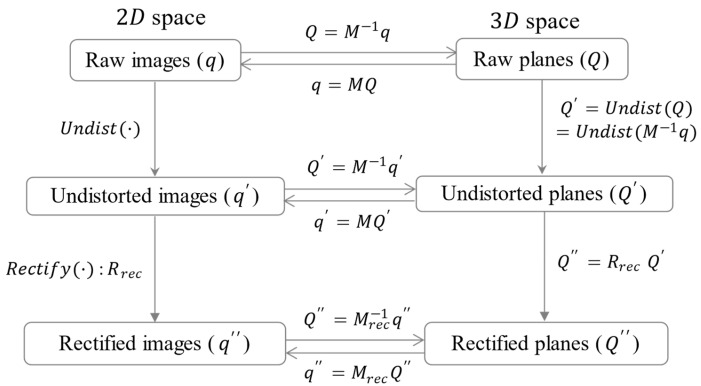
Calibration, lens-distortion correction, and rectification process diagram for a single camera.

**Figure 6 sensors-24-06100-f006:**
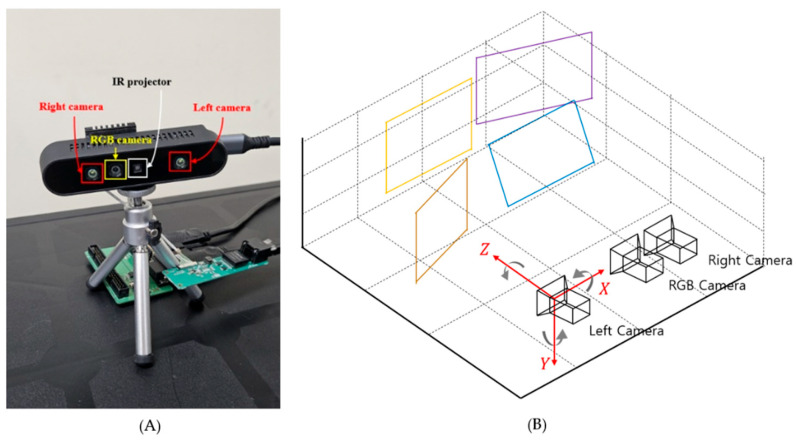
(**A**) Our stereoscopic camera with a wide field of view for distance measurement; (**B**) camera configuration and XYZ coordinates.

**Figure 7 sensors-24-06100-f007:**
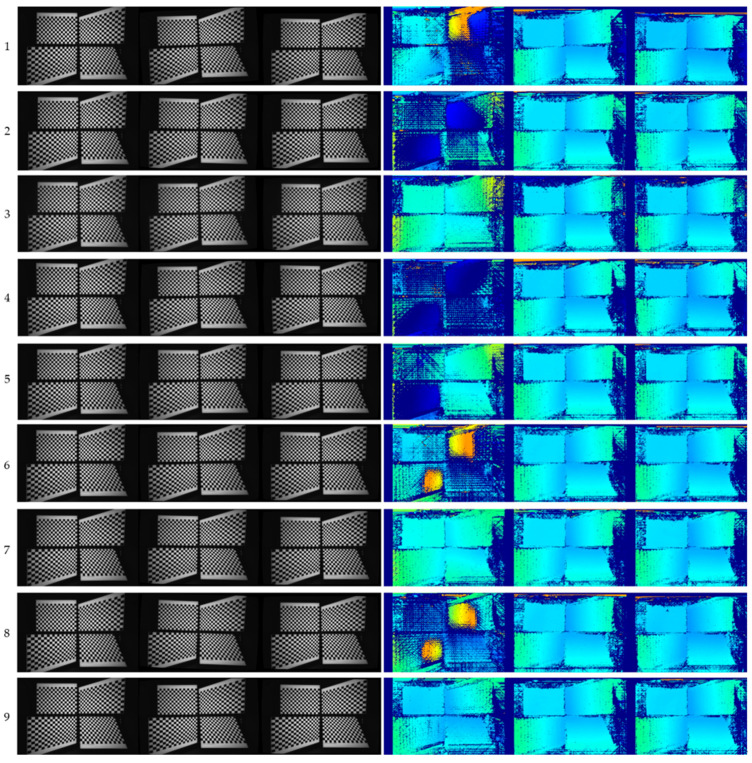
Rectification performance comparisons for each of the 19 modules. (**A**) Original left image, (**B**) rectified image using Bouguet’s method, (**C**) rectified image using the proposed method, (**D**) SGM disparity map using unrectified inputs, (**E**) SGM disparity map using Bouguet’s method, (**F**) SGM disparity map using the proposed method.

**Figure 8 sensors-24-06100-f008:**
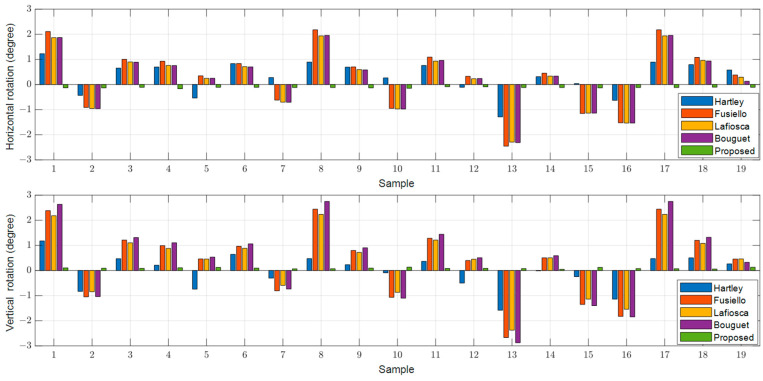
Rotated angle distributions in the rectified left images in the horizontal and vertical directions.

**Figure 9 sensors-24-06100-f009:**
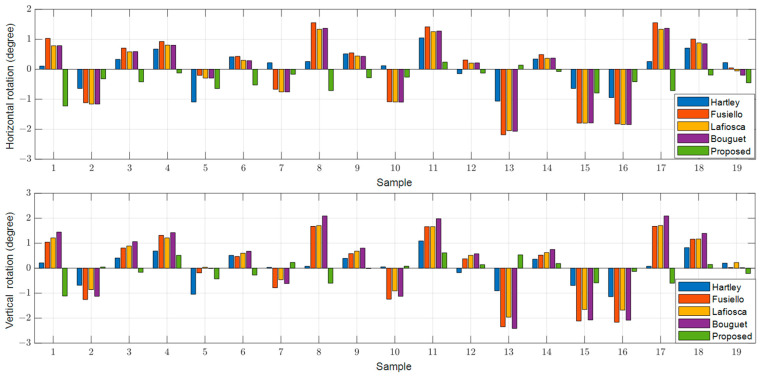
Rotated angle distributions in the rectified right images in the horizontal and vertical directions.

**Figure 10 sensors-24-06100-f010:**
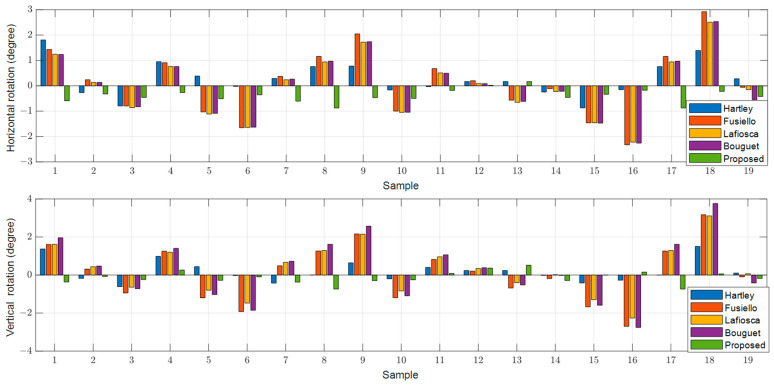
Rotated angle distributions in the rectified RGB images in the horizontal and vertical directions.

**Figure 11 sensors-24-06100-f011:**
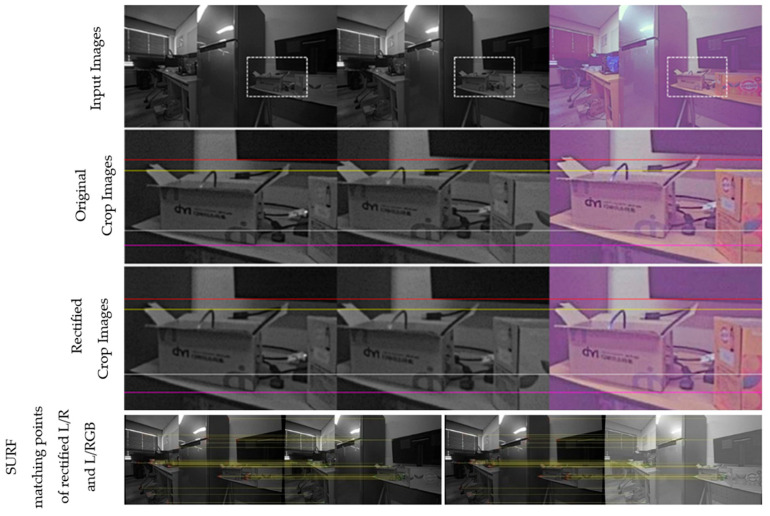
Comparison before and after rectification using the proposed method—scene 1 (left/right/RGB). Alignment errors for original images: L/R: 2.96, L/RGB: 6.75; for rectified images: L/R: 0.95, L/RGB: 0.89. Number of matching points: 33.

**Figure 12 sensors-24-06100-f012:**
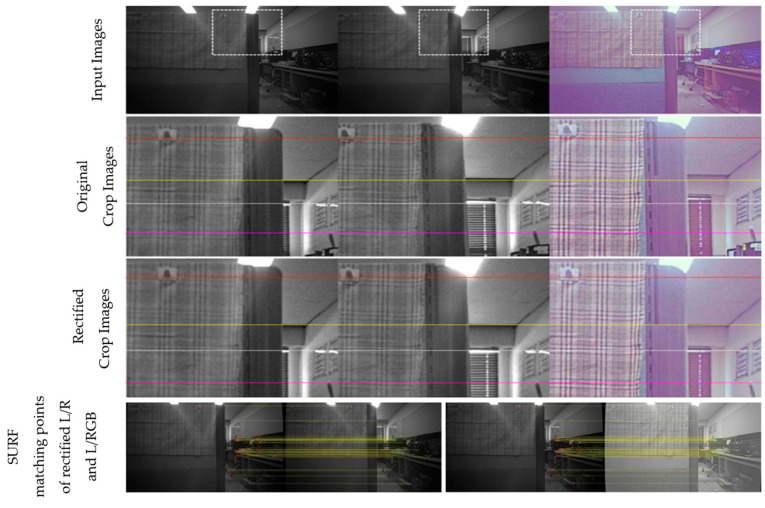
Comparison before and after rectification using the proposed method—scene 2 (left/right/RGB). Alignment errors for original images: L/R: 4.51, L/RGB: 8.21; for rectified images: L/R: 0.99, L/RGB: 1.08. Number of matching points: 34.

**Figure 13 sensors-24-06100-f013:**
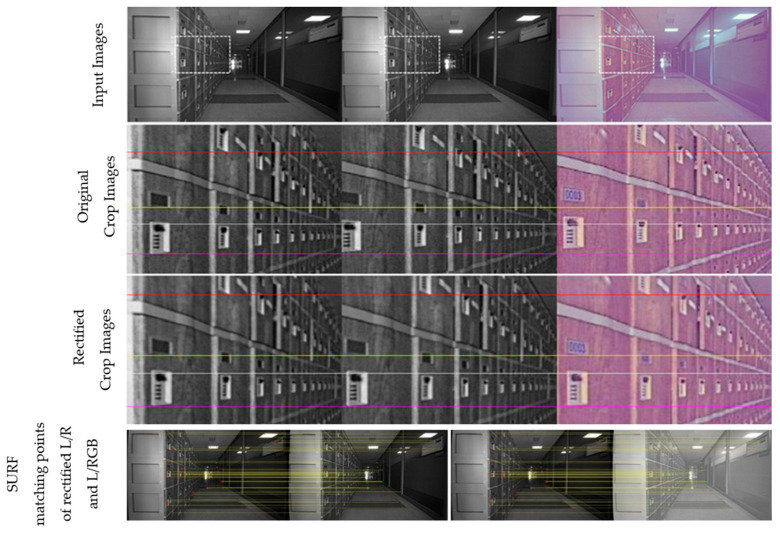
Comparison before and after rectification using the proposed method—scene 3 (left/right/RGB). Alignment errors for original images: L/R: 2.64, L/RGB: 6.36; for rectified images: L/R: 0.89, L/RGB: 0.85. Number of matching points: 46.

**Figure 14 sensors-24-06100-f014:**
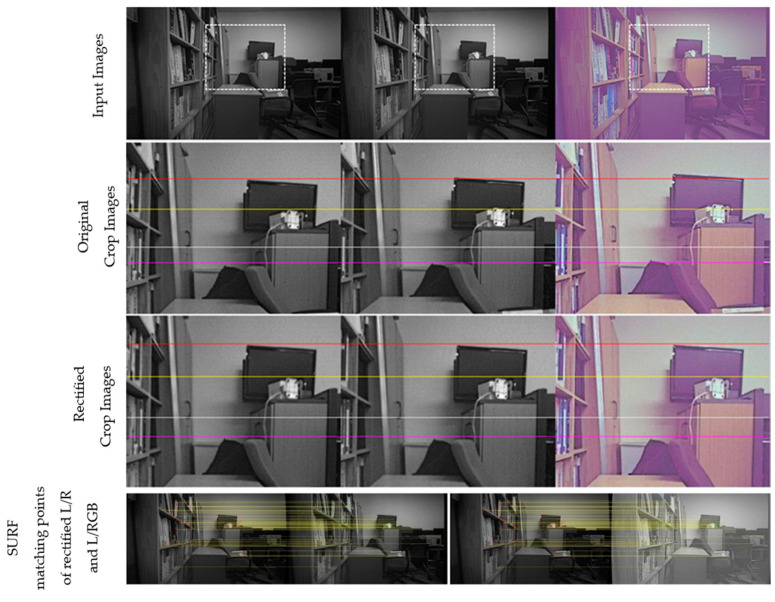
Comparison before and after rectification using the proposed method—scene 4 (left/right/RGB). Alignment errors for original images: L/R: 3.05, L/RGB: 7.49; for rectified images: L/R: 0.83, L/RGB: 1.05. Number of matching points: 53.

**Figure 15 sensors-24-06100-f015:**
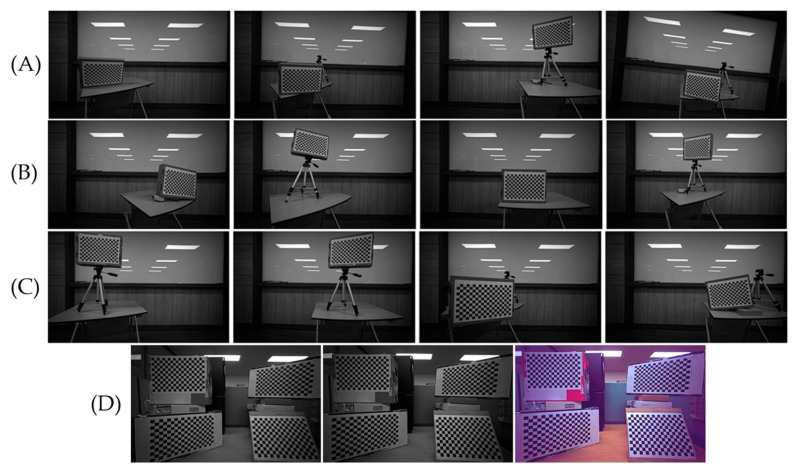
Four calibration settings for comparing multiple and single captures: (**A**) Multi-I: the first multiple capture, (**B**) Multi-II: the second multiple capture, (**C**) Multi-III: the third multiple capture, and (**D**) represents a single capture.

**Figure 16 sensors-24-06100-f016:**
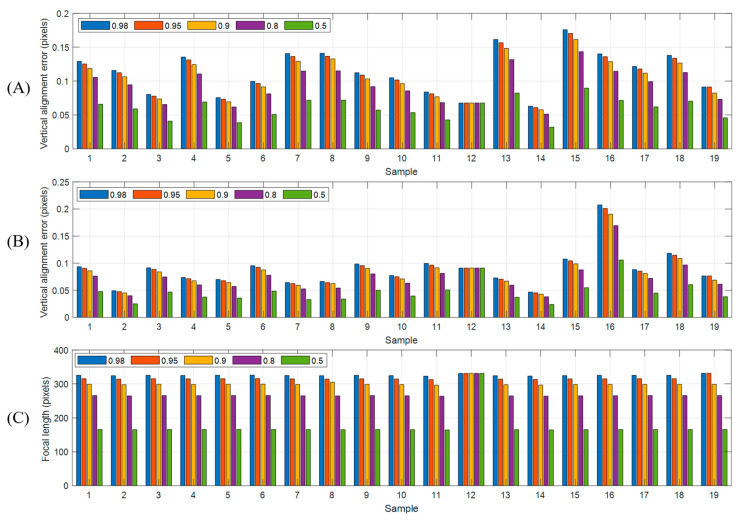
Vertical alignment errors by γ for each sample: (**A**) L/R, (**B**) L/RGB, and changes in focal length of rectified image (**C**).

**Table 1 sensors-24-06100-t001:** Comparison of $err_{v}\left(L,\;R\right)$ and $err_{v}\left(L,\;RGB\right)$ for the existing methods and the proposed method for 19 samples.

Num.	Original		Hartley [14]		Fusiello [19]		Lafiosca [20]		Bouguet [12]		Proposed	
	L/R	L/RGB	L/R	L/RGB	L/R	L/RGB	L/R	L/RGB	L/R	L/RGB	L/R	L/RGB
1	3.502	5.139	0.147	0.112	0.143	0.141	0.107	0.070	0.107	0.077	0.127	0.090
2	3.693	2.650	0.142	0.101	0.216	0.159	0.097	0.045	0.088	0.046	0.115	0.043
3	1.314	6.671	0.105	0.097	0.193	0.161	0.051	0.080	0.059	0.069	0.077	0.090
4	5.161	6.009	0.132	0.124	0.181	0.169	0.067	0.078	0.067	0.071	0.135	0.069
5	2.368	1.880	0.122	0.100	0.192	0.147	0.044	0.050	0.056	0.053	0.074	0.067
6	4.007	1.020	0.092	0.101	0.144	0.130	0.053	0.084	0.060	0.068	0.094	0.092
7	0.445	8.102	0.142	0.077	0.198	0.133	0.070	0.034	0.071	0.039	0.139	0.059
8	4.052	6.075	0.195	0.122	0.266	0.194	0.105	0.054	0.101	0.060	0.138	0.062
9	1.572	1.959	0.110	0.080	0.204	0.155	0.063	0.042	0.056	0.052	0.109	0.096
10	10.372	4.878	0.136	0.096	0.195	0.164	0.093	0.095	0.084	0.070	0.101	0.071
11	4.902	13.379	0.121	0.083	0.171	0.190	0.044	0.027	0.062	0.038	0.081	0.100
12	0.446	2.356	0.100	0.085	0.216	0.146	0.086	0.044	0.064	0.043	0.063	0.087
13	6.517	4.865	0.162	0.100	0.239	0.176	0.181	0.062	0.140	0.054	0.161	0.065
14	1.622	1.115	0.103	0.077	0.161	0.113	0.037	0.027	0.045	0.041	0.056	0.038
15	5.822	1.365	0.126	0.107	0.167	0.147	0.072	0.071	0.073	0.064	0.174	0.102
16	1.111	0.752	0.132	0.112	0.205	0.145	0.111	0.067	0.088	0.081	0.138	0.208
17	6.145	3.775	0.170	0.130	0.289	0.157	0.056	0.060	0.076	0.074	0.118	0.084
18	5.750	3.420	0.120	0.104	0.161	0.149	0.040	0.050	0.051	0.061	0.135	0.112
19	0.898	0.911	0.116	0.103	0.194	0.133	0.056	0.064	0.050	0.060	0.091	0.074
min	0.445	0.752	0.092	0.077	0.143	0.113	0.037	0.027	0.045	0.038	0.056	0.038
max	10.372	13.379	0.195	0.130	0.289	0.194	0.181	0.095	0.140	0.081	0.174	0.208
avg	3.668	4.017	0.130	0.101	0.197	0.153	0.075	0.058	0.074	0.059	0.112	0.085

**Table 2 sensors-24-06100-t002:** Relational angles and locations between L/R and L/RGB camera from calibration. Note that applying this value to the R or RGB image will transform their coordinates to match those of the L camera. Misalignments exceeding ±1 mm in the y axes are marked in red.

Num.	L/R						L/RGB					
	Relational Angle (Degree)			Relational Location (mm)			Relational Angle (Degree)			Relational Location (mm)		
	x	y	z	x	y	z	x	y	z	x	y	z
1	0.24	0.42	−1.16	−49.10	−0.92	0.06	0.111	0.084	−0.490	−36.322	−0.966	−0.095
2	−0.19	0.02	−0.22	−50.61	1.07	−0.31	−0.006	−0.203	−0.217	−38.061	−0.187	−0.104
3	−0.04	−0.62	−0.36	−50.82	−0.65	−0.44	−0.169	−0.440	−0.426	−37.803	0.572	0.081
4	−1.81	0.38	0.04	−49.35	−0.85	−0.38	−1.407	0.367	−0.103	−36.140	−0.617	0.042
5	−0.14	−0.20	−0.60	−50.68	0.18	0.11	−0.160	−0.256	−0.465	−37.325	0.732	0.082
6	−0.01	0.28	−0.42	−49.95	−0.39	−0.21	0.000	0.171	−0.262	−36.943	1.160	−0.176
7	−0.16	0.32	−0.03	−50.45	0.66	−0.38	−0.068	0.560	−0.502	−37.787	−0.303	0.602
8	−0.09	−0.38	0.10	−49.78	1.55	−0.66	0.020	−0.628	0.246	−37.083	0.152	−0.308
9	0.03	−0.14	−0.20	−50.34	−0.47	−0.64	0.411	−0.422	−0.402	−37.850	−1.388	−0.312
10	−0.08	−0.12	−0.14	−50.03	1.02	−0.41	−0.064	−0.296	−0.417	−37.759	0.737	−0.338
11	−0.04	0.17	0.36	−50.30	−1.36	0.34	0.037	0.013	−0.076	−38.085	−0.517	0.492
12	0.07	−0.05	−0.02	−51.69	−0.32	0.10	−0.107	0.094	0.125	−37.796	−0.111	−0.350
13	−0.22	−0.19	0.26	−50.94	2.04	−0.17	−0.156	−0.078	0.299	−37.358	0.419	−0.226
14	−0.03	−0.19	0.01	−50.69	−0.43	−0.35	0.059	−0.208	−0.393	−37.260	0.105	−0.241
15	−0.13	0.20	−0.72	−49.18	1.71	−0.69	−0.078	0.094	−0.211	−36.592	1.017	−0.378
16	−0.07	−0.16	−0.34	−49.76	1.75	−0.62	−0.151	−0.026	−0.071	−35.715	1.606	−1.003
17	0.11	−0.44	−0.71	−50.66	−1.42	−0.35	−0.007	−0.396	−0.854	−37.227	−0.766	−0.397
18	−0.11	−0.06	−0.08	−50.33	−0.91	−0.73	−0.017	−0.148	−0.120	−37.484	−1.993	−0.031
19	−0.06	−0.02	−0.37	−50.57	−0.02	−0.41	−0.143	−0.001	−0.343	−36.673	0.060	−0.234
min	−0.01	−0.02	0.01	−49.10	−0.02	0.06	0.000	−0.001	−0.071	−35.715	0.060	−0.031
max	−1.81	−0.62	−1.16	−51.69	2.04	−0.73	−1.407	−0.628	−0.854	−38.085	−1.993	−1.003
avg	−0.14	−0.04	−0.24	−50.28	0.12	−0.32	−0.100	−0.090	−0.246	−37.224	−0.015	−0.152

**Table 3 sensors-24-06100-t003:** Summary of rectification accuracy of checkboard corners for multiple and single captures.

Multiple Capture						Single Capture	
Multi-I		Multi-II		Multi-III			
L/R	L/RGB	L/R	L/RGB	L/R	L/RGB	L/R	L/RGB
0.047	0.037	0.070	0.102	0.195	0.201	0.106	0.094

**Table 4 sensors-24-06100-t004:** Summary of rectification accuracy in Figure 11, Figure 12, Figure 13, Figure 14 for multiple and single captures.

	Figure 11		Figure 12		Figure 13		Figure 14	
	L/R	L/RGB	L/R	L/RGB	L/R	L/RGB	L/R	L/RGB
Multi-I	0.269	0.338	0.427	0.442	0.462	0.484	0.320	0.432
Single	0.236	0.374	0.667	0.615	0.338	0.388	0.328	0.437

## Data Availability

The original contributions presented in the study are included in the article, further inquiries can be directed to the corresponding author.

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
