# Peer review of "Triple-Camera Rectification for Depth Estimation Sensor"

_sensors, 2024, doi:10.3390/s24186100_

Round 1
Reviewer 1 Report
Comments and Suggestions for Authors
The author proposes a rectification method for three cameras using a single image for depth estimation. Some questions are as follows.
1. Suggest adding references to some theories and formulas mentioned in Section 3.
2. In Section 4, how are the values of α and β chosen? According to the description of the article, it is believed that these two values should have a greater impact on the result.
3. According to Table 1, only 14 samples are listed, but the article describes 15 samples. Moreover, the average alignment error according to the values listed in Table 1 is inconsistent with the text description.
4. As the results shown in Table 1, compared to Bouguet’s method, which provides L/R stereo rectification, the L/R alignment error of the proposed method is marginally higher. Therefore, only compared with the performance of the Bouguet’s method, I think it is not enough. Bouguet’s method is a method that provides L/R stereo rectification, and its performance is better within this range than the method proposed in the article.
5. As mentioned by the author in the article, there are some methods similar to the author's work at present, and it is recommended to supplement with the same type of methods to do performance comparison.
6. After analyzing the results of Figure 12, the author mentions that “the disparity maps calculated from Bouguet’s and the proposed rectified images are quite similar. In both methods, the disparity map is enhanced to a level where the 3D geometry of the calibration chart can be reliably estimated.” Does this result indicate that the two methods perform similarly? If so, combined with the results listed in Table 1 above, Bouguet’s method performs better in some sense.
Reviewer 2 Report
Comments and Suggestions for Authors
General
- This paper addresses the limitations of existing algorithms that calibrate based on only two cameras by presenting a study on a method that simultaneously calibrates three cameras.
- The description of the method in the manuscript makes it difficult to fully understand the precise concept of the parameter estimation being performed. The optimization process is encapsulated within three general equations that merely list the parameters, but there is a lack of proposed solutions for the challenges of simultaneously optimizing a large number of parameters.
- Additionally, while the title emphasizes the use of a "single image," it should be noted that the method actually involves bundling four separate captures of a checkerboard into a single image. The advantages of using a single image are not clearly demonstrated, and the manuscript lacks sufficient validation of this approach.
- Although the motivation behind the research is commendable, the manuscript lacks detailed descriptions of the specific methods and results, as well as sufficient validation of the intermediate steps and overall outcomes. Furthermore, the absence of a discussion section contributes to the paper's lack of completeness.
Major
- There is a lack of clear definition for the vertical alignment error presented in Table 1. Additionally, while the title of Table 1 indicates average, maximum, and minimum values, only a single value is presented.
- Figures 8 to 11 qualitatively confirm the correct operation of the algorithm, but there are no quantitative metrics provided. Instead of focusing on practical application results, it might be more effective to present how each step is modified using the single checkerboard image emphasized in this study.
- A detailed explanation is needed regarding the calculation results of each parameter described in the Method section, including the errors during the calculation process. Given the large number of parameters estimated for each camera module, significant differences in convergence or errors are likely to occur, and these should be discussed.
- It is necessary to verify the accuracy of the calibration using a single image by comparing it with the results obtained from multiple captures.
- In Figure 12, it is unclear why only 10 samples were used instead of the original 15 data points. Additionally, while the Bouguet’s method is described as showing random rotations, it is not clear exactly what kind of rotations are observed. Despite these stated differences, the disparity map does not show significant variation, which needs to be examined.
- Although various discussions are relevant to this study, the manuscript lacks a separate discussion section. For example, the reasons for selecting the alpha and beta values and their effects, why the proposed method shows better rectification results for L/RGB than for L/R, and ideas for future improvements should be added to enhance the paper’s value.
Comments on the Quality of English Language-
Round 2
Reviewer 1 Report
Comments and Suggestions for Authors
The authors addressed all the comments made by me and revised the manuscript. There are still a few minor issues, such as the normalization of references in the text. In addition, perhaps due to display, the author profile is missing and needs to be supplemented.
Author Response
Comments 1: The authors addressed all the comments made by me and revised the manuscript. There are still a few minor issues, such as the normalization of references in the text. In addition, perhaps due to display, the author profile is missing and needs to be supplemented.
Response 1: Thank you for the commnet. We have accordingly revised the references in the manuscript. The references were formatted using the MDPI TeX style, and we carefully made all necessary modifications. We have also added photographs of all authors in the Author Biographies section.
Reviewer 2 Report
Comments and Suggestions for Authors
- Most of the revision comments have been addressed.
- While concerns remain regarding the convergence of parameter estimation during optimization and potential challenges in reproducing the study, the key points of the research that the paper aims to convey have been well incorporated.
Comments on the Quality of English LanguageN/A
Author Response
Comments 1: While concerns remain regarding the convergence of parameter estimation during optimization and potential challenges in reproducing the study, the key points of the research that the paper aims to convey have been well incorporated.
Response 1: Thank you for the commnet.
In MATLAB's lsqnonlin function with the "Levenberg-Marquardt" algorithm, there are control parameters such as StepTolerance, FunctionTolerance, OptimalityTolerance, MaxFunctionEvaluations , MaxIterations and that determine when to stop the optimization. All these parameters are relative and work as follows:
- StepTolerance specifies the minimum allowable change in the parameters for the algorithm to continue optimization.
- FunctionTolerance determines how small the change in the objective function value must be for the algorithm to consider that it has converged.
- OptimalityTolerance sets a threshold for the first-order optimality condition, which involves the gradient (or Jacobian) of the objective function.
The algorithm is considered to have converged and stops iterating if any one of these three criteria is met.
- MaxFunctionEvaluations sets the maximum number of function evaluations the algorithm can perform.
- MaxIterations specifies the maximum number of iterations allowed.
If either of these two limits is reached before convergence, the algorithm stops, indicating it has not converged.
The table below compares the default settings in MATLAB with our tuned version:
|   | StepTolerance | FunctionTolerance | OptimalityTolerance | MaxFunctionEvaluations | MaxIterations |
| Matlab default | 1.00E-06 | 1.00E-06 | 1.00E-06 | '200*numberOfVariables' | 4.00E+02 |
| Ours | 1.00E-25 | 1.00E-25 | 1.00E-25 | 100000 | 1.00E+05 |
The tables below show the rectification errors for both L/R and L/RGB configurations under the MATLAB default settings and our tuned settings:
| L/R | 1 | 2 | 3 | 4 | 5 | 6 | 7 | 8 | 9 | 10 | 11 | 12 | 13 | 14 | 15 | 16 | 17 | 18 | 19 | avg |
| default | 0.129 | 0.538 | 0.080 | 0.135 | 0.077 | 0.100 | 0.141 | 0.141 | 0.112 | 0.105 | 0.084 | 0.161 | 0.161 | 0.063 | 0.176 | 0.141 | 0.122 | 0.138 | 0.091 | 0.142 |
| paper | 0.129 | 0.116 | 0.080 | 0.135 | 0.076 | 0.100 | 0.141 | 0.141 | 0.112 | 0.105 | 0.084 | 0.068 | 0.161 | 0.063 | 0.176 | 0.140 | 0.122 | 0.138 | 0.091 | 0.115 |
| L/RGB | 1 | 2 | 3 | 4 | 5 | 6 | 7 | 8 | 9 | 10 | 11 | 12 | 13 | 14 | 15 | 16 | 17 | 18 | 19 | avg |
| default | 0.093 | 0.203 | 0.092 | 0.074 | 0.071 | 0.095 | 0.064 | 0.066 | 0.098 | 0.077 | 0.100 | 0.091 | 0.073 | 0.047 | 0.108 | 0.228 | 0.088 | 0.118 | 0.125 | 0.101 |
| paper | 0.093 | 0.049 | 0.092 | 0.074 | 0.070 | 0.095 | 0.064 | 0.066 | 0.098 | 0.077 | 0.100 | 0.091 | 0.073 | 0.047 | 0.108 | 0.207 | 0.088 | 0.118 | 0.076 | 0.089 |
It appears that the optimization parameters specifically affect three cases (#2, #12, and #19), while the results for the other cases remain unchanged.
In terms of convergence, both MATLAB's default settings and our custom settings ensure convergence based on our observations. However, when the MaxIterations is reduced to 100, the algorithm stops without converging in three modules (#2, #15, and #19), even though the rectification accuracy remains unchanged. This suggests that the algorithm finds an optimal solution, either globally or locally.